

# Boundary-layer water vapor profiling using differential absorption radar

Richard J. Roy, Matthew Lebsock, Luis Millán, Robert Dengler, Raquel Rodriguez Monje, Jose V. Siles, and Ken B. Cooper

Jet Propulsion Laboratory, California Institute of Technology, Pasadena, California, USA

**Correspondence:** Richard J. Roy (richard.j.roy@jpl.nasa.gov)

**Abstract.** Remote sensing of water vapor in the presence of clouds and precipitation constitutes an important observational gap in the global observing system. We present ground-based measurements using a new radar instrument operating near the 183 GHz $H_2O$ line for profiling water vapor inside of planetary-boundary-layer clouds, and develop an error model and inversion algorithm for the profile retrieval. The measurement technique exploits the strong frequency dependence of the radar beam

attenuation, or differential absorption, on the low-frequency flank of the water line in conjunction with the radar's ranging capability to acquire range-resolved humidity information. By comparing the measured differential absorption coefficient with a millimeter-wave propagation model, we retrieve humidity profiles with 200 meter resolution and typical statistical uncertainty of 0.6 $g/m^3$ out to around 2 km. The measured spectral variation of the differential absorption coefficient shows good agreement with the model, validating both the measurement method assumptions and the measurement error model. By performing

the retrieval analysis on statistically independent data sets corresponding to the same observed scene, we demonstrate the reproducibility of the measurement. An important trade-off inherent to the measurement method between retrieved humidity precision and profile resolution is discussed using an ensemble of ground-based measurements.

## 1 Introduction

In this work, we discuss the implementation of differential absorption radar (DAR) for measuring humidity profiles inside of boundary-layer clouds (Lebsock et al., 2015; Millán et al., 2016). The DAR method, which is the microwave analog of the mature differential absorption lidar (DIAL) method (Browell et al., 1979), combines the range-resolving capabilities of radar with the strong frequency dependence of atmospheric attenuation near a molecular rotational absorption line to retrieve density profiles of the absorbing gas along the line of sight. Recently, there was a demonstration of microwave integrated path

differential absorption in airborne measurements of sea surface air pressure without range resolution (Lawrence et al., 2011), utilizing the 60 GHz $O_2$ line to measure the total oxygen column. More recently, our group demonstrated a ground-based DAR for humidity sounding operating between 183 and 193 GHz (Cooper et al., 2018), with primary sensitivity to upper



tropospheric water vapor due to significant attenuation in the lower troposphere at these frequencies. That work included a comparison of differential absorption measurements with a millimeter-wave propagation model showing good agreement, and left the topics of error analysis and profile inversion for future investigation. While the 183 to 193 GHz band is attractive for DAR measurements because of the large differential absorption values achievable, transmission at frequencies between 174.8

and 191.8 GHz is prohibited due to reservation for passive-only remote sensing (NTIA, 2015). On the other hand, the 167 to 174.8 GHz band offers fewer transmission restrictions, and features lower absolute absorption, thus enabling penetration into the boundary layer from an airborne or spaceborne platform. Of course, the smaller absolute absorption is accompanied by decreased differential absorption, making the profiling capabilities of this radar coarser than the 183 to 193 GHz DAR. Furthermore, the surface returns in both cloudy and clear-sky areas make possible a DAR measurement of the total water

column.

The DAR approach has two unique aspects that complement the weakness of existing methods to remotely sense water vapor. First, because of its ranging capabilities it has precise height registration unlike passive sounding where weighting functions can encompass broad swaths of the atmosphere. Second, in contrast with other methods the DAR signal increases with increasing cloud water content and precipitation. The DAR therefore nicely complements the infrared and microwave sounding

techniques, as well as differential absorption and Raman lidar techniques that are commonly used to remotely sense water vapor from the ground (Whiteman et al., 1992; Wulfmeyer and Bösenberg, 1998; Spuler et al., 2015), with a notable airborne DIAL system being the Lidar Atmospheric Sensing Experiment (LASE) (Browell et al., 1998). Importantly, millimeter-wave transparency in clouds allows for airborne or spaceborne measurements of lower tropospheric humidity in cloudy scenes, while DIAL systems typically cannot measure inside boundary-layer clouds due to high optical thickness.

Here we present ground-based measurements using a 167 to 174.8 GHz DAR, provide in-depth error analysis with emphasis on the role of background noise power, and develop a retrieval algorithm based on performing least squares fits of a spectroscopic model to the data. The retrieved profiles constitute the first active remote sensing measurements of water vapor profiles inside of clouds, and open up possibilities for a variety of scientific studies, including investigation of in-cloud humidity heterogeneity and the coupled relationship between boundary-layer clouds and thermodynamic profiles.

## 2 Measurement basis and method

### 2.1 Differential absorption radar

The DAR technique (Lawrence et al., 2011; Millán et al., 2014; Cooper et al., 2018) utilizes range-resolved radar echoes at multiple carrier frequencies in the vicinity of a gaseous absorption line to probe the frequency-dependent optical depth between two points along the radar line-of-sight. The radar echoes, or returns, may originate from cloud hydrometeors or, in the case

of an airborne system, from the Earth's surface as well, enabling total column optical depth measurements. For closely spaced transmission frequencies near the absorption line center, the hydrometeor scattering properties vary little while the gaseous absorption exhibits strong frequency dependence. By comparing with a known propagation model, these measurements can be employed to retrieve range-resolved density profiles of the absorbing molecule. Furthermore, because of the differential nature





of the measurement, one does not require absolute calibration of the radar receiver in order to obtain absolute density values for the absorbing molecule. In the case of a calibrated receiver, both range-resolved density profiles of the absorbing molecule and microphysical properties of the reflecting medium can be retrieved.

Assuming negligible multiple scattering, the radar echo power received from a collection of scatterers filling the beam at a
distance $r$ is

$$P_e(r,f) = C(f)Z(r,f)r^{-2}e^{-2\tau(r,f)}, \tag{1}$$

where $C(f)$ includes the frequency dependence of the radar hardware (e.g. transmit power and gain), $Z(r,f)$ is the (unattenuated) reflectivity, and $\tau(r,f)$ is the one-way optical depth including contributions from gaseous and particulate extinction. Taking the ratio of powers for two different ranges $r_1$ and $r_2 = r_1 + R$ and assuming frequency-independence of the reflectivity
and particulate extinction, we find

$$\frac{P_e(r_2,f)}{P_e(r_1,f)} = \frac{Z(r_2)}{Z(r_1)}\left(\frac{r_1}{r_2}\right)^2 e^{-2\beta(r_1,r_2,f)R}, \tag{2}$$

where

$$\beta(r_1,r_2,f) = \frac{\tau(r_2,f) - \tau(r_1,f)}{R}$$
$$= \frac{1}{R}\int_{r_1}^{r_2}\left[\sum_j \rho_j(r)\kappa_j(r,f) + \beta_{\text{part}}(r)\right]dr \tag{3}$$

is the average absorption coefficient between $r_1$ and $r_2$, $\rho_j(r)$ is the density of the gas component with label $j$, $\kappa_j(r,f)$ is the corresponding mass extinction cross section, which varies with $r$ due to pressure and temperature, and $\beta_{\text{part}}(r)$ is the particulate extinction coefficient integrated over local particle size distributions.

Restricting our analysis to millimeter-wave propagation near the 183 GHz water vapor absorption line, the sum over gaseous absorption terms can be replaced by $\rho_v(r)\kappa_v(r,f) + \beta_{\text{gas,bg}}(r)$, where the subscript $v$ corresponds to water vapor and $\beta_{\text{gas,bg}}$
is the background gas absorption coefficient due to all other components, which is assumed to be frequency independent. Assuming that pressure and temperature vary slowly compared to the length scale $R$, we can therefore write equation (3) as

$$\beta(r_1,r_2,f) = \bar{\rho}_v(r_1,r_2)\kappa_v(f) + \bar{\beta}_{\text{gas,bg}}(r_1,r_2) + \bar{\beta}_{\text{part}}(r_1,r_2), \tag{4}$$

where the overbar symbol implies taking the mean value between $r_1$ and $r_2$. Thus, we see that measuring the frequency-dependent contribution to the optical depth between $r_1$ and $r_2$ reveals the average water vapor density given the known absorp-
tion line shape $\kappa_v(f)$.

Figure 1 shows the frequency dependence of the gaseous absorption coefficient $\rho_v\kappa_v(f) + \beta_{\text{gas,bg}}$ in the vicinity of the 183 GHz water vapor line for $P = 1000$ mbar, $T = 285$ K, and $\rho_v = 10$ g/m³. For this work, we utilize the millimeter-wave propagation model from the EOS Microwave Limb Sounder (Read et al., 2004). The 167 to 174.8 GHz transmission band is highlighted in green, as well as shown in the inset to figure 1, revealing a differential absorption coefficient of 3 dB/km for





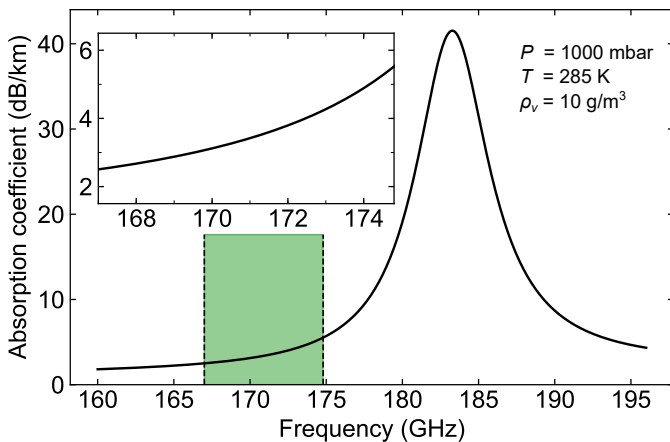

**Figure 1.** Gaseous absorption coefficient (one-way) calculated using the model from Read et al. (2004) and the parameters listed in the figure. The green shaded region and inset highlight the 167 to 174.8 GHz transmission band for this work.

10 g/m$^3$ of water vapor. Important to the validity of this DAR method is the dominance of gaseous differential absorption over particulate differential absorption, since we assume that $\beta_{\text{part}}$ is frequency independent. To investigate this, we estimate the frequency dependence of $\beta_{\text{part}}$ by integrating the extinction cross section for liquid water Mie spheres over particle size distributions consistent with rain. We do this for characteristic hydrometeor diameters in the range of 0.3 to 3 mm, and vary

the total particle concentration in order to maintain a rainfall rate of 10 mm/h. From these calculations we find a maximum differential absorption from particulate extinction of 0.01 dB/km, which is roughly two orders of magnitude smaller than that from water vapor for typical boundary-layer humidity values. Therefore, we are confident that the frequency dependence in our measured values of $\beta(r_1, r_2, f)$ is due to water vapor absorption, and systematic effects from particulate scattering are negligible.

**2.2   FMCW radar basics and instrument details**

Due to the lower transmit power as compared to conventional radar systems at lower frequencies, the 170 GHz radar is operated in a frequency-modulated continuous-wave (FMCW) mode, which can offer increased sensitivity relative to a pulsed system with the same power because the transmitter is always on. The basic principle of FMCW radar is outlined in figure 2. The transmitted signal is frequency-modulated with a linear chirp waveform of bandwidth $\Delta F_{\text{chirp}}$ and duration $T_{\text{chirp}}$. After

scattering off of a target at a distance $r$ from the radar, the received chirp is delayed in time by an amount $2r/c$, where $c$ is the speed of light, leading to a fixed frequency offset of $\delta f = 2\Delta F_{\text{chirp}} r/(cT_{\text{chirp}})$ relative to the transmitted frequency chirp. By downconverting the received signal using the transmitted frequency $f(t)$ shifted by 5 MHz for convenient amplification and detection, the fixed frequency offset between transmitted and received chirps is converted into a constant frequency signal in the intermediate frequency (IF) stage. Signal processing techniques are then used to convert the IF time-domain signal to a

range-resolved power spectrum. In the IF power spectrum, the zero-range point is located at 5 MHz and the echo power from a





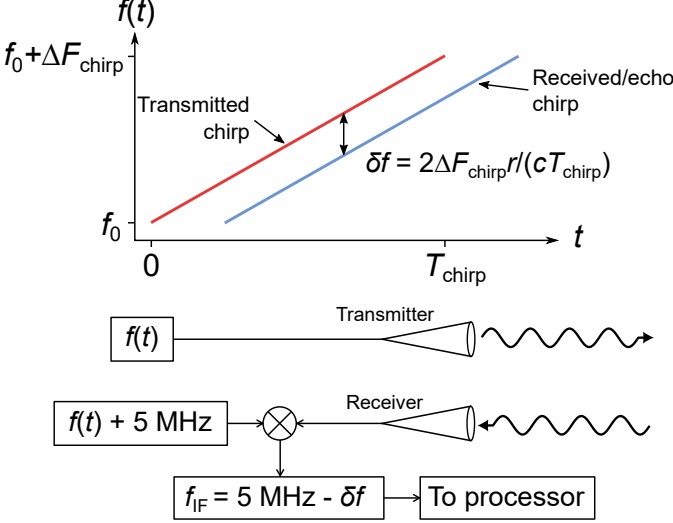

**Figure 2.** Basic FMCW radar schematic. See Section 2.2 for discussion.

range $R$ is located at $f_{\mathrm{IF}}(r) = 5\,\mathrm{MHz} \pm \delta f(r)$, where the positive(negative) sign applies for decreasing(increasing) frequency chirps.

Our system utilizes state-of-the-art millimeter-wave components designed at the Jet Propulsion Laboratory (JPL) and builds on years of FMCW radar development for security and planetary science applications (Cooper et al., 2011, 2017). The archi-

tecture is similar to that presented in an earlier work (Cooper et al., 2018) which demonstrated the DAR technique between 183 and 193 GHz, but modified to transmit in the less restricted 167 to 174.8 GHz band, to perform narrow-bandwidth frequency chirps, and to provide a 5 MHz offset of the zero-range radar signal from zero frequency within the IF band. The IF offset is helpful for future calibrated power measurements because of various effects that inhibit accurate power estimation near zero frequency. The radar has an average transmit power of 140 mW, is outfitted with a 6 cm primary aperture with corresponding

gain of 40 dB, and uses a frequency chirp of bandwidth $\Delta F_{\mathrm{chirp}} = 60\,\mathrm{MHz}$ and duration $T_{\mathrm{chirp}} = 1\,\mathrm{ms}$, resulting in a range resolution of $\Delta r = 2.5\,\mathrm{m}$. Planned future upgrades to the system will include the installation of a 60 cm primary aperture and corresponding 20 dB increase in gain, as well as an increase in transmit power by a factor of 4.

To process the downconverted radar signal, we first sample it using an analog-to-digital converter (ADC) with a sampling frequency of 20 MHz for the 1 ms duration of the chirp. Then we apply a Hanning window in the time domain before performing

a fast Fourier transform (FFT) to obtain the range-resolved power spectrum. Application of the Hanning window reduces sidelobes from bright targets as well as the large transmit/receive leakage signal that is always present at zero-range. For the radar parameters listed above, the corresponding conversion factor from IF frequency to target range is $\delta f(r)/r = 400\,\mathrm{kHz/km}$.





### 2.3 Power measurement uncertainty

The starting point for assessing the achievable precision in humidity using DAR measurements is the statistical uncertainty of the radar power measurements themselves. Until this point, we have ignored the role of background noise power in the radar spectrum, which is an important factor in any realistic receiver. In general, the noise power within a given radar range bin $P_n$ is

proportional to the sum of the receiver noise temperature and the antenna temperature, which itself is proportional to the scene brightness temperature. By considering the simultaneous coherent detection of noise ($P_n$) and radar echo ($P_e$) power, one can show that the statistical uncertainty of the *detected* power, $P_d = P_e + P_n$, is given by (see Appendix A)

$$\sigma_d = \frac{1}{\sqrt{N_p}} \left( P_e^2 + 2P_e P_n + P_n^2 \right)^{1/2}, \tag{5}$$

where $N_p$ is the number of radar pulses transmitted.

In order to accurately determine the frequency dependent optical depth between two range bins, it is critical to obtain a separate measurement of the background noise power in the absence of radar echoes and subtract this off of $P_d$. To see why this is, consider equation (2) with the left hand side replaced by $P_d(r_2, f)/P_d(r_1, f)$, which is equivalent to interpreting the detected power as the true echo power, set $Z(r_2) = Z(r_1)$ for simplicity, and consider the limit $P_e \ll P_n$ (i.e. $P_d \to P_n$). In this case we would find that $\exp(-2\beta(r_1, r_2, f)R) \to 1$ regardless of the actual value of $P_e$, and thus would incorrectly estimate a

vanishing water vapor density, when in fact it is the echo power which has vanished. Similarly, for modest values of the signal-to-noise ratio $\text{SNR} \equiv P_e/P_n$, this would lead to a systematic underestimate of the true humidity. Therefore, after subtracting the separate noise power measurement from $P_d$ we obtain a measurement of $P_e$ with total uncertainty $\sigma_e = (\sigma_d^2 + \sigma_n^2)^{1/2}$, where $\sigma_n = P_n/\sqrt{N_p}$ is the noise power measurement uncertainty (see equation (5) with $P_e = 0$). The relative uncertainty in the measured echo power is therefore

$$\frac{\sigma_e}{P_e} = \frac{1}{\sqrt{N_p}} \left( 1 + \frac{2}{\text{SNR}} + \frac{2}{\text{SNR}^2} \right)^{1/2}. \tag{6}$$

As will be discussed in Section 3, the range dimension is purposefully oversampled in our measurements, allowing us to decrease the statistical power uncertainty at a given range by averaging $N_b$ adjacent range bins. The resulting relative power uncertainty is given by

$$\frac{\sigma_e}{P_e} = \frac{\xi(N_b)}{\sqrt{N_p N_b}} \left( 1 + \frac{2}{\text{SNR}} + \frac{2}{\text{SNR}^2} \right)^{1/2}, \tag{7}$$

where $\xi(N_b) \geq 1$ is a factor of order unity accounting for covariances between adjacent range bins that arise due to applying a window function to the time-domain radar signal before transforming to Fourier space. For the Hanning window used in this work, this function is given by $\xi(N_b) = \left( 1 + \frac{N_b - 1}{N_b} \frac{8}{9} \right)^{1/2}$.

### 2.4 Inversion algorithm for profile retrieval

Under the simplifying assumptions introduced in the Section 2.1, and assuming that pressure and temperature are known as a

function of range, the inverse problem to retrieve humidity can be solved directly. The implications of the latter assumption





are explored in Appendix C. To invert the radar spectra, we consider a set of measured echo powers $P_e(r_i, f_j)$ for ranges $\{r_1, r_2, ..., r_m\}$ and transmission frequencies $\{f_1, f_2, ..., f_{N_f}\}$, where $r_{i+1} - r_i = \Delta r$ is the radar range resolution. We note that in most circumstances we employ a retrieval step size $R$ that is larger than $\Delta r$, since, as we'll show below, the precision in our retrieved humidity scales favorably with total optical depth and hence with increasing $R$. Then, given a step size such that

$R = r_{i+S} - r_i$ for some integer $S$, we form the frequency-dependent measured quantity

$$\gamma_i(f_j) = -\frac{1}{2R} \ln\left[ \left(\frac{r_{i+S}}{r_i}\right)^2 \frac{P_e(r_{i+S}, f_j)}{P_e(r_i, f_j)} \right] \tag{8}$$

for each starting range $r_i$. From equation (2), we see that we can extract the average humidity between $r_i$ and $r_{i+S}$ by performing a least squares fit of the function

$$\hat{\gamma}(f) = \bar{\rho}\kappa(f) + B \tag{9}$$

to the measurements for each $i$, where $B$ is a frequency-independent offset containing information about dry air gaseous absorption, particulate extinction, and the relative reflectivity of the two ranges in question. We drop the $v$ subscript in the above equation for simplicity of notation. The resulting humidity estimates $\{\bar{\rho}_1, \bar{\rho}_2, ..., \bar{\rho}_{m-S}\}$ have a corresponding range axis $\{\bar{r}_1, \bar{r}_2, ..., \bar{r}_{m-S}\}$ where $\bar{r}_i = (r_i + r_{i+S})/2$, and have associated uncertainties determined from the fitting procedure.

Using standard error propagation, the estimated uncertainty in the measured quantity $\gamma_i(f_j)$ defined in equation (8) is

$$\sigma_{\gamma_i}(f_j) = \frac{1}{2R} \left[ \left( \left.\frac{\sigma_e}{P_e}\right|_{r_{i+S}, f_j} \right)^2 + \left( \left.\frac{\sigma_e}{P_e}\right|_{r_i, f_j} \right)^2 \right]^{1/2}. \tag{10}$$

In order to derive a simple analytical expression for the relative uncertainty in the retrieved humidity, we restrict ourselves for the moment to considering two transmission frequencies, $f_1$ and $f_2$. In this case, we can combine equations (2), (4), and (8) to obtain the humidity directly,

$$\bar{\rho}(\bar{r}_i) = [\kappa(f_2) - \kappa(f_1)]^{-1} [\gamma_i(f_2) - \gamma_i(f_1)], \tag{11}$$

with the associated relative uncertainty

$$\left.\frac{\sigma_{\bar{\rho}}}{\bar{\rho}}\right|_{\bar{r}_i} = \frac{1}{2\Delta\tau} \left[ \sum_{j=1,2} \left( \left( \left.\frac{\sigma_e}{P_e}\right|_{r_{i+S}, f_j} \right)^2 + \left( \left.\frac{\sigma_e}{P_e}\right|_{r_i, f_j} \right)^2 \right) \right]^{1/2}, \tag{12}$$

where $\Delta\tau = [\kappa(f_2) - \kappa(f_1)]\bar{\rho}(\bar{r}_i)R$ is the differential optical depth for $f_1$ and $f_2$ between range bins $r_i$ and $r_{i+S}$. Equation (12) reveals that there are three linked quantities determining the sensitivity of the system: (1) the magnitude of the DAR signal quantified by $\Delta\tau$, (2) the statistical uncertainty of the power measurements given by the quadrature sum of relative errors in

equation (12), and (3) the relative uncertainty in the derived value for the humidity. Thus, given a set of measured echo powers and a specific value for the humidity, there is a trade-off between spatial resolution of the retrieval and relative uncertainty in the humidity estimate. These ideas will be explored in Section 4.





An important and subtle point regarding the uncertainty in the measured quantity $\gamma_i(f_j)$ is that equation (10) relies on a Taylor expansion in the relative error $\sigma_e/P_e$, and therefore is only valid for measurements with SNR above some critical value that depends on the number of measurements $N_p$. Because there is no closed-form expression for the probability distribution function (PDF) of $\gamma_i(f_j)$, we resort to a Monte Carlo analysis, which is described in Appendix B, to generate numerically the relevant PDFs for the parameters used in this work. From this analysis, we find that for $N_p = 2000$ pulses and $N_b = 11$ averaged bins, the Taylor expansion method is accurate for measurements with SNR > 0.1.

We note here that it is typical of differential absorption systems to utilize only two frequencies: one on-line, and one off-line. However, in this work we are concerned with validating both the spectroscopic model used and the radar hardware itself, which could be subject to unknown frequency-dependent systematic effects. The regression approach discussed above thus provides for a robust comparison of the measured frequency dependence $\gamma_i(f_j)$ with the model $\hat{\gamma}(f)$, while a two-frequency approach would mask inconsistencies between measurements and model, or systematic hardware effects, since the two free parameters $\bar{\rho}$ and $B$ are fully determined given two frequency points. Furthermore, a distributed set of frequencies allows for the possibility of extending retrievals deeper in range for moist atmospheres, as frequencies closer to the line center will be attenuated more strongly, and can be excluded from the fits described above when the critical SNR value is reached.

## 3 Boundary-layer measurements and analysis

### 3.1 Radar characteristics, spectra, and filtering

In this section we report on measurements performed at JPL on March 15, 2018 using the proof-of-concept differential absorption radar described in Section 2.2. For these measurements, we implement a new signal processing technique for real-time noise floor characterization, utilizing a triangle-wave frequency chirp (i.e. bidirectional) instead of a sawtooth-wave (i.e. unidirectional). According to FMCW radar principles, the echo spectrum switches from residing on the low to the high-frequency side of the zero-range signal (i.e. 5 MHz) for increasing and decreasing linear frequency chirps, respectively. As shown in figure 3(a), this fast switching of the chirp direction alternately exposes the noise floor on each side of the zero-range point within the IF band, and provides accurate and nearly continuous estimation of the system noise power and the passive signal corresponding to the scene brightness temperature at each frequency bin. This technique is especially advantageous for airborne/spaceborne applications, as the brightness temperature of the observed scene can change on fast timescales due to different surface types (e.g. ocean vs. land) and from the presence or absence of clouds.

Figure 3 showcases a few aspects of a single ground-based DAR measurement, for which the conditions were light drizzle and a cloud located a few hundred meters off the ground. For all the field measurements discussed in this work, we acquire $N_p = 2000$ pulses for each of 12 frequencies equally spaced between 167 and 174.8 GHz, with the radar positioned just inside a building, pointing at $30°$ elevation. The experimental sequence is as follows: first, we perform 40 frequency chirps at a given transmission frequency before switching to another frequency, which takes 1 ms. The received signal is downconverted to baseband, digitized in an analog-to-digital converter (ADC), and processed in real-time as described in Section 2.2. We achieve a system duty cycle of > 90%, resulting in a total measurement/observation time of $\approx 25$ seconds.





**Figure 3.** DAR measurement spectra. (a) The bidirectional frequency chirp technique provides for accurate, real-time characterization of the background noise floor within the radar's IF band, with no loss of measurement duty cycle. Here the detected power spectrum for $f_j = 167$ GHz is shown. The IF frequency to range conversion factor is 400 kHz/km. (b) Echo power spectra normalized to their value at 100 m for the 12 transmission frequencies. The large variability in the signals near 1.4 km indicates the system reaching the noise floor. (c) Echo power spectra after averaging $N_b = 11$ adjacent bins, and filtered for points with SNR > 0.1. (d) Measurement relative error (blue circles) for all traces in (c) compared with the statistical model (equation (7), dashed black line).





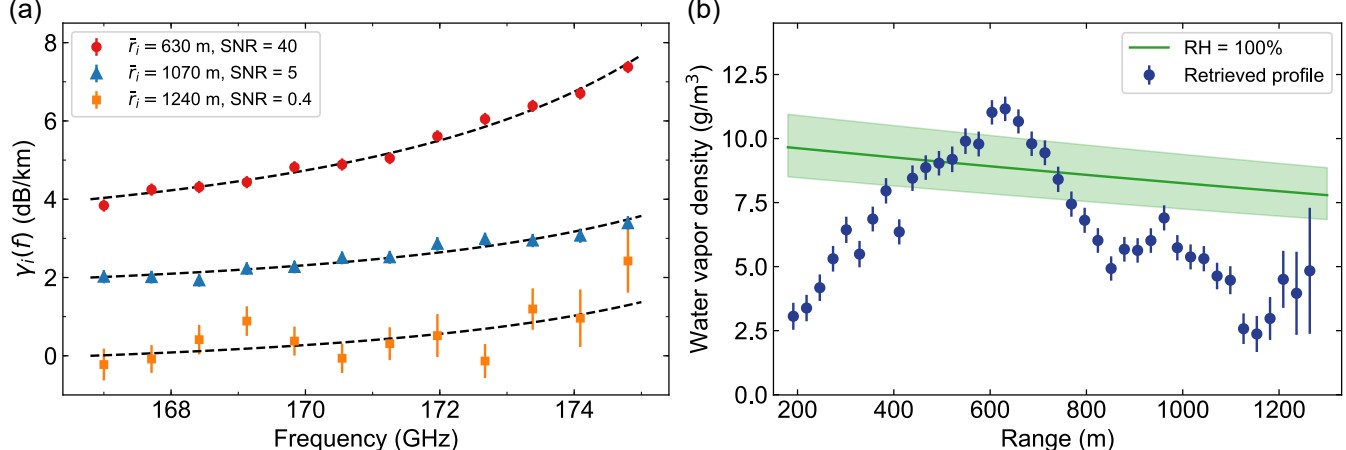

**Figure 4.** Water vapor profile retrieval for DAR spectra from figure 3. (a) Three examples of least-squares fits of the millimeter-wave propagation model to DAR measurements. Artificial offsets are imposed in order to plot all three on the same graph. (b) The retrieved profile exhibits roughly constant absolute humidity error until SNR $\approx 10$ (1 km). See Sections 2.4 and 3.2 for retrieval details. The green line shows the saturated water vapor density range dependence using a near-surface temperature of $11\,^{\circ}$C and lapse rate of $6\,^{\circ}$C/km. The shaded regions correspond to deviations of $\pm 2\,^{\circ}$C.

By subtracting the respective noise floors from the increasing and decreasing frequency chirp measurements (figure 3(a)), and subsequently combining the mirrored spectra, we obtain our estimate of the echo power spectra. In figure 3(b), we plot the echo power spectra scaled by $r^2$ for the 12 transmission frequencies before bin averaging, which reveals the range dependence of the quantity $Z(r)\exp(-2\tau(r,f))$. Each spectrum is normalized to its value at 100 meters. Thus, we observe the differential

absorption due to water vapor directly from the spreading of the spectra with increasing range, where for a particular range, the plotted values increase monotonically with decreasing transmit frequency. After averaging the quantity $r_i^2 P_e(r_i, f_j)$ within a swath of size $N_b = 11$, we filter the spectra based on the Monte Carlo analysis in Appendix B, keeping only those points with SNR $> 0.1$, and are left with the smoothed profiles shown in figure 3(c). Figure 3(d) shows the relative error in the binned ($N_b = 11$) echo power measurement (blue circles) plotted against the measured SNR for all 12 frequencies. The measured

values agree very well with those predicted by equation (7) (black dashed line), indicating that our statistical model based on speckle noise, which underlies the Monte Carlo simulations implemented in this work, is accurate.

### 3.2 Water vapor profile retrieval

Using the averaged, filtered spectra in figure 3(c), we proceed towards retrieving the water vapor density profile using the procedure outlined in Section 2.4. For the profiles presented in this section, we utilize a retrieval step size of $R = 200$ m.

Beginning with an initial range of $r_1 = 100$ m, we form the 12 quantities $\gamma_i(f_j)$ for each starting $r_i$ in the set $\{r_1, r_2, ..., r_{m-S}\}$, and perform a least-squares fit of the function $\hat{\gamma}(f)$ to the data at each range point. Note that the retrieved water vapor density $\bar{\rho}_i$ is related only to the difference between the value of the fitted function at 174.8 and 167 GHz, while the offset is related to



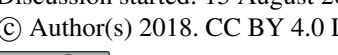



**Figure 5.** Retrievals for different cloud and precipitation conditions. (a) Averaged DAR power spectra ($N_b = 11$) for light rain near the surface, with a cloud extending from 1 to 2 km. (b) Retrieved humidity profiles for two independent data sets corresponding to the same scene from (a). (c) Averaged DAR power spectra ($N_b = 11$) for heavy precipitation near the surface with strong particulate extinction. (d) Independent retrievals from two data sets for scene in (c).

particulate extinction and hydrometeor reflectivity, and is disregarded in this work. The pressure and temperature dependence of the absorption line shape is included in the fitting model using reported values at the surface from a nearby weather station, and assuming an exponential pressure profile with a scale height of 7.5 km and a temperature lapse rate of $6\,°C/km$. We note that for the relatively short vertical extents of the profiles from these ground measurements (e.g. $1.4\sin 30°$ km for figures 3 and 4), the retrieved $\bar{\rho}$ values are quite insensitive to the assumed thermodynamic profiles (see Appendix C).

An important element of the DAR technique in general is utilizing an accurate model for the absorption line shape. Examples of line shape fits to the data are shown in figure 4(a) for three different values of SNR, with arbitrary offsets imposed on the three traces to permit simultaneous plotting. To assign SNR values to these points, we compute the mean SNR for the 12 frequencies at $r_i$ and $r_{i+S}$, and use the smaller of the two. Clearly the millimeter-wave model accurately captures the frequency dependence





of the measurements, which is supported quantitatively by the typical reduced-chi-square values of $\chi^2_{\mathrm{red}} \approx 1$ for these fits. The retrieved water vapor density profile is shown in figure 4(b), where the range $\bar{r}_i$ assigned to each fitted value $\bar{\rho}_i$ is the midpoint of $r_i$ and $r_{i+S}$. Also plotted here is an estimate of the saturation vapor density given our lapse rate assumption. This profile is consistent with a cloud base between 400-600 meters and shows qualitatively good agreement with the expectation that

the relative humidity is approximately 100% in liquid cloud layers. Note that because the retrieved values correspond to the mean humidity between $r_i$ and $r_{i+S}$, we effectively retrieve the profile convolved with a box of size $R$ (200 m here). For this retrieval, the absolute humidity errors lie between 0.55 and 0.60 g/m$^3$ until around 1 km (SNR $\approx 10$), where the error steadily increases until the final retrieval point at 1.25 km with $\sigma_{\bar{\rho}} = 2.9$ g/m$^3$. The value of $\sigma_{\bar{\rho}}$ in the high-SNR regime (i.e. the first 1 km) remains roughly constant, even though $\bar{\rho}$ varies by a factor of 3, since the absolute humidity error is independent of the

humidity itself, and depends only on the differential mass extinction cross section $\kappa(174.8 \, \mathrm{GHz}) - \kappa(167 \, \mathrm{GHz})$, the retrieval step size $R$, and the power measurement uncertainty (see equation (12)).

Though we do not have independent, coincident water vapor profile measurements with which to validate the accuracy of the retrieval, we can investigate repeatability and consistency of this DAR method by performing the retrieval on coincident, independent DAR measurements of the same exact scene. To do so, we acquire $N_p = 4000$ pulses at each frequency with a total

measurement time of 50 seconds, and parse the data into two groups of $N_p = 2000$ pulses both spanning the full 50 seconds. The results are shown in figure 5, where we also present measurements of different cloud and precipitation scenarios than that presented in figures 3 and 4. In figure 5, (a) and (b) correspond to light rain at the surface with a cloud boundary at 1 km range, and (c) and (d) to heavy rain at the surface with strong particulate extinction. The retrievals from the two independent sample sets in both cases agree quite well, which showcases the reproducibility of the measurement and indicates that the estimated

humidity error accurately captures the sample scatter.

## 4   Discussion of retrieval precision and resolution

Given a measured range-resolved echo power spectrum, what retrieval range resolution can we achieve for a specified minimum retrieval precision? As discussed briefly in Section 2.4, the relative error in the retrieved humidity $\sigma_{\bar{\rho}}/\bar{\rho}$ (see equation 12) for a given power measurement uncertainty varies inversely with the differential optical depth, and thus depends on both the

retrieval step size $R$ (i.e. retrieval resolution) used *and* the absolute value of the humidity $\bar{\rho}$. To investigate this trade-off between retrieval resolution and absolute humidity, we utilize 18 separate DAR measurements with identical experimental parameters, all averaged and filtered in the same way discussed in Section 3.1. We then perform the retrieval discussed in the previous section for many different retrieval step sizes in the range $55 \, \mathrm{m} \leq R \leq 578 \, \mathrm{m}$, allowing us to acquire statistics for a large range of differential optical depth values. We first analyze the results by binning the aggregate data set based on relative

error in the retrieved humidity for the intervals $(0, 0.1]$, $(0.1, 0.2]$, and $(0.2, 0.5]$. We bin three different variables according to their associated value of $\sigma_{\bar{\rho}}/\bar{\rho}$: the one-way differential absorption coefficient $\Delta\beta_{174.8-167} = \bar{\rho}(\kappa(174.8 \, \mathrm{GHz}) - \kappa(167 \, \mathrm{GHz}))$, the retrieval step size $R$, and a measure of the total power measurement uncertainty (and therefore SNR) for all frequencies $f_j$



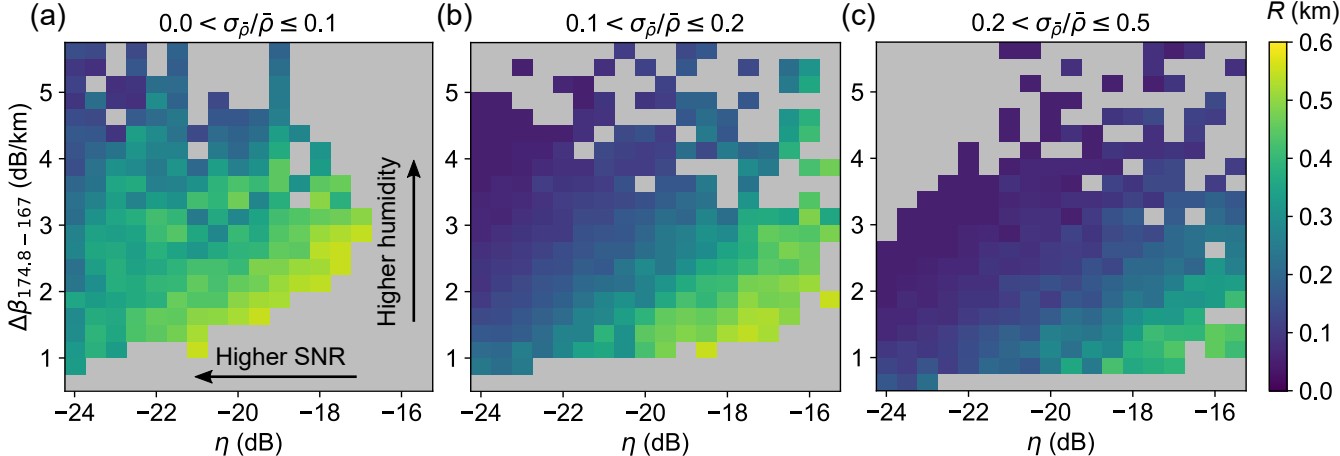

**Figure 6.** Statistics of retrieval range resolution $R$ as a function of power measurement uncertainty $\eta$ and local differential absorption coefficient $\Delta\beta_{174.8-167}$. The 2D plots in (a), (b), and (c) correspond to binning these three quantities according to the retrieved humidity relative error in the intervals listed.

at $r_i$ and $r_{i+S}$,

$$\eta_i = \frac{1}{N_f}\left[\sum_{j=1}^{N_f}\left(\left(\frac{\sigma_e}{P_e}\bigg|_{r_{i+S},f_j}\right)^2 + \left(\frac{\sigma_e}{P_e}\bigg|_{r_i,f_j}\right)^2\right)\right]^{1/2}, \tag{13}$$

which reduces to $\eta = \eta_0/\sqrt{N_f/2}$ in the case where all relative errors are equal $\sigma_e/P_e = \eta_0$. Furthermore, because of the power-law-like dependence of $\sigma_e/P_e$ on SNR, we express the quantity $\eta$ in dB in order to capture the full dynamic range sampled by the data set. Note that lower values of $\eta$ correspond to higher-SNR measurements. For example, $\eta = -24.2$ dB for SNR $\gg 1$, and $\eta = -20.8$ dB in the case where all $2N_f$ measurements have SNR $= 1$.

Figure 6 shows the results of this binning procedure in the form of 2D plots of average retrieval step size as a function of $\Delta\beta_{174.8-167}$ and $\eta$. The gray regions correspond to parts of the parameter space that were not realized in the field measurements. For all three humidity relative error intervals, we clearly see the trade-off in retrieval resolution (i.e. step size $R$) that must be made for smaller humidity values (i.e. smaller $\Delta\beta$) in order to maintain a given retrieval precision. For comparison, recall from Section 2.1 that for $\rho_v = 10$ g/m$^3$, $P = 1000$ mbar, and $T = 285K$, the differential absorption coefficient is 3 dB/km. In future work we will incorporate these uncertainty considerations into an algorithm that has adaptive range resolution based on both the inherent signal (i.e. humidity) and the measurement noise.

# 5 Conclusions

A proof-of-concept humidity-profiling DAR operating between 167 and 174.8 GHz has been constructed and tested from the ground. The instrument builds on progress made in an earlier version operating between 183 and 193 GHz (Cooper et al.,





2018), and employs a new signal processing technique for performing real-time noise power spectrum characterization and subtraction, providing for higher accuracy measurements of the radar echo power. A new direct inversion algorithm for retrieving humidity based on least squares fits to a spectroscopic model is applied to the measured echo power spectra, showing close agreement between the measurement and model frequency dependence. The humidity profiles retrieved from two statistically

independent measurement sets of the exact same scene are in close agreement, highlighting the reproducibility of the method. The uncertainties in the power measurements, which in part determine the retrieved humidity uncertainty, agree very well with a statistical model based on radar speckle noise that incorporates the effects of background noise subtraction and down sampling, or binning, of the measured spectra.

Development of an operational airborne 167-174.8 GHz DAR is currently in progress, which will include an additional 20

dB of antenna gain and a factor of 4 increase in transmit power. Important future steps for this instrument include validation of the measurement accuracy using coincident measurements of humidity, pressure, and temperature (e.g. from radiosondes), and eventually testing from an airborne platform. Specifically, the surface returns while measuring from an airborne platform will be investigated for retrieval of total column water within the boundary layer. A more significant augmentation of the system could include the addition of passive radiometric channels near the 183 GHz line. This would allow for continuous

measurement of vertical humidity profiles when transitioning between clear sky and cloudy areas, and opens the possibility to study biases in the humidity retrieved from radiometric measurements that are caused by scattering and emission from clouds.

## Appendix A:  Error in detected power

In this appendix, we derive the expression for the detected power uncertainty within a single radar range bin in the presence of background noise (equation (5)). To do so, we begin by assuming that all targets within the scattering volume are randomly

distributed, leading to the well-known Rayleigh fading model for the received echo signal (Ulaby et al., 1982). In the context of FMCW radar, we then consider the received complex electric field amplitude $E_i$ corresponding to the $i^{\text{th}}$ frequency bin in the FFT spectrum, where we only consider the polarization direction that couples into the radar receiver. Within the Rayleigh fading model, it is shown that for $E_i = E_{0,i} e^{i\phi_i}$, the modulus of the field amplitude $E_{0,i}$ is normally distributed with zero mean and standard deviation $\sigma_E$, and the phase $\phi_i$ is uniformly distributed over the interval $[0, 2\pi]$. Alternatively, we can write

the corresponding voltage in the receiver as $V_{e,i} = a_i + ib_i$, where $a_i$ and $b_i$ are uncorrelated and are both normally distributed with zero mean. Then, from the expression converting electric field to power, $P_{e,i} = |V_{e,i}|^2 = \alpha|E_i|^2$, we find the probability distribution function for the received echo power

$$p(P_{e,i} \geq 0) = \frac{1}{\langle P_{e,i} \rangle} e^{-P_{e,i}/\langle P_{e,i} \rangle}, \tag{A1}$$

where the mean equals the variance and is given by $\langle P_{e,i} \rangle = 2\alpha\sigma_E^2$, $\alpha$ is a field-to-voltage conversion factor for the radar, and

$p(P_{e,i} < 0) = 0$. Furthermore, we find that $\langle a_i^2 \rangle = \langle b_i^2 \rangle = \langle P_{e,i} \rangle/2$. Though not proven here, the Rayleigh fading model also shows that the expectation value of the received power from $N$ randomly distributed targets is the sum of the expectation values of the individual target echo powers.



Similarly, one can show that Gaussian white noise in the radar signal, which comes from both the scene brightness temperature and the radar electronics, results in a noise voltage within the $i^{\text{th}}$ frequency bin of the FFT spectrum with Fourier coefficient $V_{n,i} = c_i + id_i$, where $\langle c_i \rangle = \langle d_i \rangle = 0$ and $\langle c_i^2 \rangle = \langle d_i^2 \rangle = \langle P_{n,i} \rangle / 2$. We proceed towards deriving equation (5) by considering the coherent detection of both the radar echo and noise signals. In this case, the detected voltage signal in the Fourier domain

within the $i^{\text{th}}$ range bin is $V_{d,i} = V_{e,i} + V_{n,i}$, and the detected power is $P_{d,i} = |V_{d,i}|^2$. Using the expectation values listed above, it is easy to show that

$$\langle P_{d,i} \rangle = \langle P_{e,i} \rangle + \langle P_{n,i} \rangle \tag{A2}$$

and

$$\text{Var}(P_{d,i}) = (\langle P_{e,i} \rangle + \langle P_{n,i} \rangle)^2 . \tag{A3}$$

Therefore, we recover equation (5) by computing the standard error for $N$ independent measurements, $\sigma_{d,i}^2 = \text{Var}(P_{d,i})/N$.

**Appendix B:  Monte Carlo Analysis**

As discussed in Section 2.3, subtracting off the noise power contribution to the detected power $P_d$ is critical for accurate humidity estimation. However, for low values of SNR, we clearly expect the result $\langle P_e \rangle = \langle P_d \rangle - \langle P_n \rangle$ to be negative some of the time due to finite sampling, where "$\langle \cdots \rangle$" denotes the sample average. This is non-physical. Therefore, in order to account

for these finite sampling effects and the potential breakdown of standard error propagation when $\sigma_e / P_e$ is not small, we employ a Monte Carlo simulation of the DAR measurement. The PDF for the echo power received from randomly distributed hydrometeor targets within a single range bin is given by equation A1. In order to generate random samples of the radar spectrum for transmission frequency $f_j$, we begin with an idealized spectrum $\langle P_e(r_i, f_j) \rangle$ for which we set $C(f) = Z(r,f) = 1$ and $\rho_v(r) = \text{constant}$, and sample the distribution (equation (A1)) at each range bin $r_i$. We then perform a fast Fourier transform

(FFT) to obtain the corresponding time-domain radar signal, add the effects of background noise using Gaussian white noise, and apply a Hanning window. Taking the inverse FFT thus supplies a single random realization of a measured $P_d$ spectrum. For the simulated spectra used in this work, we generate $N_p = 2000$ radar pulses with range resolution $\Delta r = 2.5$ m and average them to realize a single radar measurement. These values for $N_p$ and $\Delta r$ are the same parameters utilized in the field measurements presented in Section 3. We generate 10,000 averaged spectra for both $P_d$ and $P_n$, giving 10,000 random

realizations of the echo power measurement. For these simulations, we use $f_j = 167$ GHz and $\rho_v = 7.4$ g/m$^3$.

Our aim is to utilize the Monte Carlo simulations to inform where the Taylor expansion method for error propagation breaks down in our estimation of $\sigma_{\gamma_i}(f_j)$, and thus provide a criterion for filtering our measurements. To do so, we fix $N_b = 11$ and the step size $S = 10$ (i.e. 275 meters) and compute the mean and standard deviation of the Monte Carlo probability distribution for the two-way transmission between $r_i$ and $r_{i+S}$ for each $r_i$. Figure B1 shows the results, where we plot the Monte Carlo

mean value divided by the *a priori* two-way transmission used to generate the Monte Carlo results, as a function of the SNR at $r_{i+S}$. The solid green lines represent the SNR-dependent errors predicted from equation (7). There are two notable





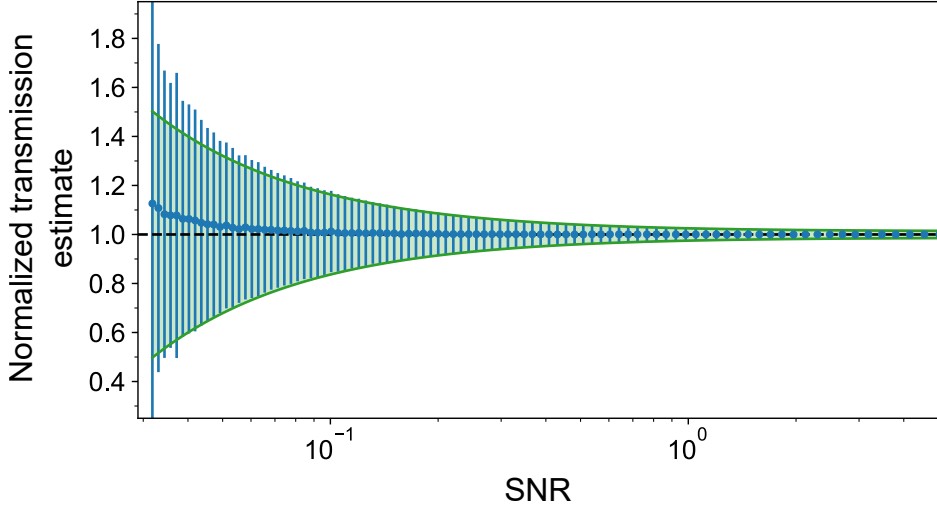

**Figure B1.** Statistics of low-SNR estimation of two-way transmission. The blue data points represent the means and standard deviations of the Monte Carlo probability distributions, while the green lines represent the error calculated using equation (7). For SNR < 0.1, the systematic bias of the Monte Carlo mean and underestimation of the true error by equation (7) imply that measurements in this region should be disregarded. In order to access lower values of SNR, one must increase the number of pulses $N_p$ for each measurement.

deviations that arise for SNR values below 0.1: (1) The error estimated using the standard error propagation formalism begins underestimating the true standard deviation calculated using the Monte Carlo ensemble, and (2) the mean of the Monte-Carlo-generated distribution systematically overestimates the true two-way transmission. We note here that this point of departure between the naive error propagation estimate and that from the Monte Carlo distributions does not depend on $N_b$ or $S$, but is determined by the number of independent pulses $N_p$ used to realize a single radar measurement. From these simulations, we conclude that the standard error propagation model is sufficient for SNR > 0.1. Therefore, after down sampling the measured spectra with $N_b = 11$, we eliminate all measured values with SNR < 0.1, as described in Section 3.1.

## Appendix C: Retrieval dependence on assumed pressure and temperature values

To assess the dependence of the retrieved humidity on temperature and pressure, we will consider again the case of the two-frequency measurement, using transmission frequencies $f_1 = 167$ GHz and $f_2 = 174.8$ GHz. Then, for a given starting range $r_i$ and step size $R$, we use the measured quantities $\gamma_i(f_1)$ and $\gamma_i(f_2)$ to solve for the mean humidity between the two ranges,

$$\bar{\rho}_i = \frac{\gamma_i(f_2) - \gamma_i(f_1)}{\kappa(f_2, P, T) - \kappa(f_1, P, T)} = \frac{\gamma_i(f_2) - \gamma_i(f_1)}{\Delta\kappa(P, T)}, \tag{C1}$$

where we now explicitly write $\kappa$ as a function of temperature $T$ and pressure $P$, and we've defined the differential mass extinction cross section $\Delta\kappa(P, T)$ for these two frequencies. Given reference values of $P_0 = 1000$ mbar and $T_0 = 285$ K, and





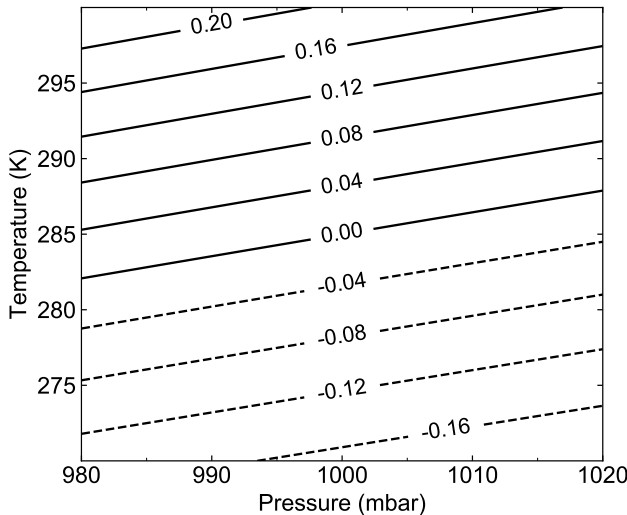

**Figure C1.** Humidity error for assumed temperature $T_0 = 285$ K and pressure $P_0 = 1000$ mbar versus actual values $T$ and $P$.

corresponding retrieved humidity $\bar{\rho}_{i,0}$, we calculate the error in our humidity estimate for different conditions $P$ and $T$ as

$$\frac{\bar{\rho}_i - \bar{\rho}_{i,0}}{\bar{\rho}_{i,0}} = \frac{\Delta\kappa(P_0,T_0)}{\Delta\kappa(P,T)} - 1. \tag{C2}$$

Figure C1 shows the humidity error for pressure deviations of $\pm 20$ mbar and temperature deviations of $\pm 15$ K. Here we see that the retrieved humidity is very weakly dependent on the assumed pressure, and only accrues an error of 10% for a temperature

5   deviation of about 8 K.

*Acknowledgements.* This research was supported by NASA's Earth Science Technology Office under the Instrument Incubator Program, and was carried out at the Jet Propulsion Laboratory (JPL), California Institute of Technology, Pasadena, CA, USA, under contract with the National Aeronautics and Space Administration. R. Roy's research was supported by an appointment to the NASA Postdoctoral Program at JPL, administered by Universities Space Research Association under contract with NASA.





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
