# Peer review of "Boundary-layer water vapor profiling using differential absorption radar"

_Atmospheric Measurement Techniques, 2018_

## Referee Comment (RC1) · Anonymous Referee #2 · 7 Sep 2018

The paper describes novel ground-based measurements performed by a DAR in the 183 water vapour absorption bands and a retrieval methodology to extract water vapour profiles from them. The paper is generally clear, well written and well presented.

I have some comments that I would like to be addressed by the authors.

1) In The statement at line 7 and 8 in the abstract you should clearly state that this is obtained in conditions of high SNR. Also it is driven by the range of your frequency within the absorption line, this should be mentioned otherwise the reader may generalize this conclusion erroneously.

2) Line 14 page 2: the authors should mention the obvious caveat of attenuation in reducing the SNR (too much water content/rain drives the signal below sensitivity).

3) Line 4 page3: it would be beneficial to discuss when the assumption of negligible multiple scattering is negligible or refer to previous literature.

4) Line 4-7 page 4: I am not fully convinced by this maximum differential absorption from particulate extinction of 0.01 dB/km. I haven't tried a specific computation but liquid cloud extinction is proportional to $1/\lambda$. So (assuming that the changes in refractive indices are negligible) a change of roughly 3% in lambda should correspond to a change of 38 dB/km/(g/m$^3$), which means that a deep cumulus cloud with 3 g/m$^3$ could produce 0.08 dB/km (an order of magnitude larger than quoted).

5) Line 10 page 5: What is the rationale for using a $\Delta F_{chirp}$ of 60 MHz and thus a range resolution of 2.5 m (with the obvious need of averaging later on for improving the SNR?)? Why not using a smaller bandwidth in first place?

6)Line 11-12 page7: I do not see the need of dropping the v subscript on $\kappa$, I would recommend to keep it for clarity (otherwise the reader may think it is the total extinction).

7)Generally in literature SNR values are stated in dB. In Fig.4 and its discussion you use linear units. Fig6 is also confusing to me, why using an obscure value like eta in the x-axis instead of using the SNR itself?

8) Fig5: it could help the reader to have a double y axis with the plot of the relative humidity and its uncertainty as well.

9) Fig5: A couple of points at low and far ranges from the two independent datasets in the bottom right panel seem to disagree, any comment?

10) Fig6: apart from the selection of the x-axis I struggle in extracting information from this figure. Why not doing a contour plot of $\sigma_\rho/\rho$ using SNR vs rho e.g. for 100 200 and 400m integration? Anyhow I would ask the authors to try to rethink the figure and present it in a more understandable way.

---

## Referee Comment (RC2) · Anonymous Referee #1 · 14 Sep 2018

This paper discusses the theoretical framework and preliminary results for in cloud absolute humidity retrievals using a new differential absorption radar technique at 183 GHz water vapor absorption line. Although mature at the optical frequencies, this is the first demonstration of range resolved differential absorption profiles of an atmospheric constituent at the microwave frequencies. The paper is generally well written and clearly organized with adequate theoretical basis to support preliminary measurements. Addressing a few general comments may strengthen the paper and provide insight into future applications of this new measurement technique.

1. Page 2, Line 9: Several references to column water vapor are made throughout the paper, this being the first one. Although the surface echo (or cloud for ground based measurements) may be exploited to directly measure the column water vapor, no discussion is presented on the challenges associated with this measurement, specifically, to what accuracy the differential power ratio between the different sounding frequencies need to be measured. Back of the envelope calculations show a relative error of 10-4 is required in the relative transmitted power ratio, which is certainly a difficult task. A brief discussion (somewhere in the paper) on the column measurement requirements would be beneficial.

2. Page 3, Line 1: Please clarify that the column measurements do require absolute calibration.

3. Page 3, Line 4: An assumption is made that the effects of multiple scattering are negligible on the received echo within clouds. This subject is not mentioned again in the paper. It is unclear that this assumption is valid and is highly dependent on the cloud water and ice particle size distributions. At the optical frequencies, lack of quantitative knowledge of the multiple scattering limits the utility of the received signal within clouds. The effects of multiple scattering become significantly larger as the beam propagates deeper into the cloud. This effect has been quantified at the microwave frequencies such as Cloud Sat and should be more completely addressed in this paper. A discussion on the impacts of multiple scattering on the humidity retrievals for different cloud particle size distributions and viewing geometries (distance to scattering target (ground vs airborne vs space) should be presented.

4. Page 4, Lines 11-13. It should be noted that comparison on sensitivity between pulsed and FMCW is dependent on background signal levels. In high background levels with the FMCW IF bandwidth compared to the background within the gate width of a pulsed system, the advantage quickly diminishes.

5. Page 5, Line 10. The advantage of selecting such a high chirp frequency (60 MHz) is not clear, especially when the DAR retrievals are done over an equivalent bandwidth of  $\sim$  2-3 MHz. Please clarify on why the higher chirp frequency was selected.

6. Page 5, Line 16. A linear chirp results in side lobes in the power spectra which can
contaminate the signal from the main lobe. Please discuss the logic behind choosing a linear chirp instead of a non-linear chirp such as one with a Gaussian frequency distribution which would result in a Gaussian response in the time domain. A plot showing the power spectra (and resulting side lobes) from a bright scatterer would be beneficial to the reader.

7. Page 5, Figure 2. Update the figure to accurately represent the bi-directional chirp discussed in the text

8. A single table describing the system parameters would be good in section 2.2. Of particular interest is the antenna beam width and spatial side lobes.

9. Page 8, First paragraph. Please clarify why background subtraction is done in the Fourier domain and not in the time domain.

---

## Referee Comment (RC3) · G. Mace (Referee) · 19 Sep 2018

This manuscript presents an innovative proof of concept for profiling water vapor in boundary layer clouds. Such measurements are difficult to impossible to make in situ and totally out of reach with remote sensing. I have several relative minor issues that the authors should address.

1. At what point will multiple scattering become a limiting factor? At these high frequencies and the typical optical depths of shallow cumulus - perhaps with coexisting precipitation - it seems that multiple scattering may be an issue.

2. Will the accuracy be sufficient to measure realistic supersaturations in cumulus updrafts? It seems that from a science perspective such knowledge is key. Then

combining this instrument with more traditional radars and lidars, one could examine aerosol cloud interaction problems by knowing the cloud droplet number concentration and humidity near cloud base where aerosol populations become activated. Additional science applications could examining the entrainment processes near cloud top where dry tropospheric air is mixed into the marine boundary layer. It seems as though the accuracy required for these topics might push the limitations of the technology.

3. For the topics identified in point 2, validation with radiosondes would be inadequate. Would current in situ technology for measuring water vapor allow for validation of the technique?

---

## Author Comment (AC1) · 22 Oct 2018

Dear reviewer #1,

Thank you for your comments and suggestions regarding our manuscript. Listed below are our itemized responses, with the original comment/question displayed in italics.

1. *Page 2, Line 9: Several references to column water vapor are made throughout the paper, this being the first one. Although the surface echo (or cloud for ground based measurements) may be exploited to directly measure the column water vapor, no discussion is presented on the challenges associated with this*

[Figure]

*measurement, specifically, to what accuracy the differential power ratio between the different sounding frequencies need to be measured. Back of the envelope calculations show a relative error of 10⁻⁴ is required in the relative transmitted power ratio, which is certainly a difficult task. A brief discussion (somewhere in the paper) on the column measurement requirements would be beneficial.*

Indeed, we mention multiple times the capability of an airborne (or spaceborne) version of this instrument to measure the total water column using surface returns. However, this is not a focus of this paper, which is concerned with profiling within boundary-layer clouds. Thus, we do not deem it appropriate to discuss the technical or systematic details pertaining to such a measurement.

Furthermore, it is not clear how the reviewer arrived at their back of the envelope calculation. If the gas extinction cross section $\kappa(f)$ doesn't change appreciably in the part of the atmosphere where the majority of water vapor resides, the total column water vapor (TCWV) for surface returns from two frequencies is

$$TCWV = \frac{1}{2\Delta\kappa} \left[ \ln\left(\frac{P_e(R_s, f_1)}{P_e(R_s, f_2)}\right) + \ln\left(\frac{C(f_2)}{C(f_1)}\right) + \ln\left(\frac{\sigma_0(f_2)}{\sigma_0(f_1)}\right) \right]. \tag{1}$$

Here $\sigma_0(f)$ is the surface cross section, $R_s$ is the distance to the surface, $[TCWV] = $ kg/m$^2$, and all other variables are as in the manuscript. The uncertainty from the first term on the right hand side corresponds to radar speckle, and thus decreases (in relative error) as the square root of the number of pulses. The main systematic concern, therefore, is the second term, related to radar calibration. Defining $\alpha = C(f_2)/C(f_1)$, the relative error in the retrieved value is therefore

$$\frac{\sigma_{TCWV}}{TCWV} = \frac{1}{2\tau}\frac{\delta\alpha}{\alpha}. \tag{2}$$

So, if we have an average of 5 g/m$^3$ of water vapor in the lowest 5 km of the atmosphere, we already have $\tau = 1.5$ for $f_1 = 167$ GHz and $f_2 = 174.8$ GHz.

Thus, a 5% relative error in the retrieved TCWV corresponds to an error in $\alpha$ of 15%. This is a significantly less demanding level of accuracy than that proposed by the reviewer. An identical argument holds for uncertainties in the differential surface cross section.

2. *Page 3, Line 1: Please clarify that the column measurements do require absolute calibration.*

   As is evident from the above analysis, absolute calibration of the radar system is **not** required, but rather only relative calibration at the two frequencies. For the same reasons discussed above, we don't feel this information should be included in the main paper, since no total column measurements are discussed.

3. *Page 3, Line 4: An assumption is made that the effects of multiple scattering are negligible on the received echo within clouds. This subject is not mentioned again in the paper. It is unclear that this assumption is valid and is highly dependent on the cloud water and ice particle size distributions. At the optical frequencies, lack of quantitative knowledge of the multiple scattering limits the utility of the received signal within clouds. The effects of multiple scattering become significantly larger as the beam propagates deeper into the cloud. This effect has been quantified at the microwave frequencies such as Cloud Sat and should be more completely addressed in this paper. A discussion on the impacts of multiple scattering on the humidity retrievals for different cloud particle size distributions and viewing geometries (distance to scattering target (ground vs airborne vs space) should be presented.*

   While multiple scattering effects are a primary concern of any spaceborne millimeter-wave radar, we disagree that discussion of multiple scattering need be a prominent part of the paper, which focuses on near-range ground based testing. However, for completeness in this response we provide a brief quantitative treatment of when this could become a problem. The degree of multiple scattering within a cloudy volume depends on the ratio $\chi = X/\ell_t$, where $X$ is the radar beam footprint at the range of interest and $\ell_t = \beta_s^{-1}(1 - \tilde{\omega}g)^{-1}$ is the *transport* mean free path, which is different from the scattering mean free path $\ell_s = \beta_S^{-1}$ (see R. Hogan, *J. Atmos. Sci.*, 65, 2008). Here $\beta_s$, $\tilde{\omega}$, and $g$ are the scattering coefficient, single-scattering albedo, and asymmetry parameter, respectively, integrated over the drop size distribution (DSD). Multiple scattering effects become important when $\chi$ is of order unity or larger.

The first attached figure shows the dependence of $\chi$ on the characteristic drop diameter of a DSD for clouds and rain at a range of 1 km. The system utilizes a 6 cm primary aperture with a 10 dB taper, corresponding to a far-field 3 dB antenna full width of 1.9 degrees. The scattering parameters are integrated over a modified gamma distribution of the form

$$N(D) = \frac{N_0}{\Gamma(\nu)D_n} \left(\frac{D}{D_n}\right)^{\nu-1} e^{-D/D_n}, \tag{3}$$

where $N_0$ is the peak number concentration, $D_n$ is the characteristic diameter, and $\nu$ is the shape parameter. Here, we use $\nu = 1$ for rain and 4 for cloud. Furthermore, we implement a parametrization of $N_0$ as a function of $D_n$, which has been shown to better match observations than e.g. Marshall-Palmer (see Abel and Boutle, Q. J. R. Meteorol. Soc., 2012). This parametrization determines the rain rate for a given $D_n$. For clouds, we fix the liquid water content (LWC) to 1 g/m$^3$. Clearly, from the figure we see that multiple scattering is not an issue (i.e. $\chi \ll 1$) for the measurements presented in this work.

To determine when multiple scattering is an issue from a spaceborne platform, it is necessary to use a more realistic antenna size for such a system. Accordingly, the second figure attached shows that same plot for a 1 meter aperture and 10 dB taper, this time for a range of 400 km. In this case, we see that there is a range of diameters for which $\chi$ is of order unity. Furthermore, one can simply scale

the values of $\chi$ for clouds linearly to consider LWC values different than 1 g/m$^3$. Multiple scattering will thus be an important consideration from a spaceborne platform. However, the modest values of $\chi$ mean that the effects are not so deleterious as to render information retrieval impossible, as in the case of in-cloud lidar measurements.

4. *Page 4, Lines 11-13. It should be noted that comparison on sensitivity between pulsed and FMCW is dependent on background signal levels. In high background levels with the FMCW IF bandwidth compared to the background within the gate width of a pulsed system, the advantage quickly diminishes.*

   We do not understand the reviewer's point here, and ask them to clarify if our explanation here doesn't suffice. In short, the background signal level, specifically meaning noise power $P_n$ within a single range bin, is only a function of noise temperature $T_n$ and integration time $\tau$, with $P_n = k_B T_n / \tau$. For all classes of radar, the signal-to-noise ratio is given by $SNR = P_e \tau / k_B T_n$, where $\tau$ corresponds to the pulse width. Of course, for conventional pulsed radar, one must use very short pulses to achieve reasonable range resolution (e.g. $\tau = 3.3$ $\mu$s for CloudSat), while for chirped-pulsed and FMCW radar, ranging is related to the chirp bandwidth, not pulse duration. Thus, using the CloudSat pulse repetition frequency of roughly 4 kHz, one can achieve the same sensitivity using an FMCW system with 3.3 $\mu$s $\times$ 4 kHz $\approx 1\%$ of the transmit power.

5. *Page 5, Line 10. The advantage of selecting such a high chirp frequency (60 MHz) is not clear, especially when the DAR retrievals are done over an equivalent bandwidth of $\approx$ 2-3 MHz. Please clarify on why the higher chirp frequency was selected.*

   The choice of chirp bandwidth involves a compromise between acquiring more statistically independent measurements (i.e. larger chirp bandwidth) within a given volume, which is advantageous for high-SNR targets where uncertainty

is limited by radar speckle, and averaging down the noise within that same volume (i.e. smaller chirp bandwidth) for low-SNR targets. In this work, we chose to purposefully oversample the range dimension with a radar resolution of 2.5 meters in order to achieve low power measurement uncertainty for our desired profile resolution of 27.5 meters. A sentence has been added in the text to clarify this point.

6. *Page 5, Line 16. A linear chirp results in side lobes in the power spectra which can contaminate the signal from the main lobe. Please discuss the logic behind choosing a linear chirp instead of a non-linear chirp such as one with a Gaussian frequency distribution which would result in a Gaussian response in the time domain. A plot showing the power spectra (and resulting side lobes) from a bright scatterer would be beneficial to the reader.*

   In short, the linear chirp is not responsible for side lobes. Side lobes in the range dimension for FMCW and chirped-pulsed radar result from the Fourier transform of the finite duration pulse. Indeed, we do limit side lobes during our digital signal processing step by applying a Hanning window to the time-domain signal before taking an FFT. See page 5 line 14 of the original manuscript. As is well known, application of a Hanning window reduces the first (and strongest) side lobe to -32 dB below the main lobe, thus removing any concern that side lobe effects contaminate adjacent radar signals.

7. *Page 5, Figure 2. Update the figure to accurately represent the bi-directional chirp discussed in the text.*

   The figure has been updated to show both directions of the chirp.

8. *A single table describing the system parameters would be good in section 2.2. Of particular interest is the antenna beam width and spatial side lobes.*

   We have added important properties of the beam profile into a new table in section 2.2.

9. *Page 8, First paragraph. Please clarify why background subtraction is done in the Fourier domain and not in the time domain.*

It is impossible to subtract Gaussian white noise in the time domain; it is the noise *power* which must be subtracted (related to the variance of the white noise in the time domain). This is done using the power spectral density in Fourier space. To see this, we consider the downconverted radar signal in the intermediate frequency (IF) band resulting from a single target volume of distributed scatterers at range $r_0$. The time domain signal is of the form $s_d(t) = s_e(t) + s_n(t) = A_e \cos(2\pi f_{IF}(r_0)t + \phi_e) + s_n(t)$. Here $A_e$ is the peak signal voltage corresponding to the target echo and $s_n(t)$ represents white noise, which is a random variable with zero mean, and is uncorrelated in time (i.e. $\langle s_n(t_1)s_n(t_2)\rangle \propto \delta(t_1 - t_2)$). Since the noise signal voltage at a given IF frequency originates from a large number of uncorrelated sources, the statistics of each Fourier component of $s_n(t)$ are identical to those within the Rayleigh fading model (i.e. speckle statistics). Clearly we can't subtract $s_n(t)$ in the time-domain signal above, since it is impossible to *simultaneously* measure $s_d$ and $s_n$. Furthermore, since the term of interest to us (the echo voltage term) contains a randomly fluctuating phase ($\phi_e$) from pulse to pulse, we can only measure the variance of the echo voltage amplitude (i.e. the power), as the mean value vanishes. For these reasons, one works with radar signals in the form of power spectral densities.

**Multiple scattering effects at 170 GHz for 6 cm aperture with 10 dB taper**

[Figure: plot of $\chi(r = 1\ \text{km})$ versus $D_n$ (m). Vertical axis ranges from $10^{-6}$ to $10^{-1}$. Horizontal axis ranges from about $10^{-5}$ to $10^{-3}$. Dashed vertical lines labeled 0.1 mm/h, 1, 10, 100. Two curves: Cloud (LWC = 1 g/m³) and Rain.]

Cloud (LWC = 1 g/m$^3$)
Rain

**Fig. 1.** Multiple scattering parameter dependence on DSD characteristic diameter for cloud and rain at a range of 1 km using a 6 cm primary aperture. See item #3 above for more details.

**Multiple scattering effects at 170 GHz for 1 m aperture with 10 dB taper**

Legend:
- Cloud (LWC = 1 g/m$^3$)
- Rain

**Fig. 2.** Same as figure 1, except for the case of a spaceborne G-band radar with a 1 meter primary aperture and a range of 400 km.

---

## Author Comment (AC2) · 22 Oct 2018

Dear reviewer #2,

Thank you for your comments and suggestions regarding our manuscript. Listed below are our itemized responses, with the original comment/question displayed in italics.

1. *In The statement at line 7 and 8 in the abstract you should clearly state that this is obtained in conditions of high SNR. Also it is driven by the range of your frequency within the absorption line, this should be mentioned otherwise the reader may generalize this conclusion erroneously.*

[Figure]

The abstract has been updated to provide this clarification regarding the quoted water vapor uncertainty.

2. *Line 14 page 2: the authors should mention the obvious caveat of attenuation in reducing the SNR (too much water content/rain drives the signal below sensitivity).*

   A sentence has been added to clarify that increased reflectivity due to increased cloud water content has an associated increase in attenuation.

3. *Line 4 page 3: it would be beneficial to discuss when the assumption of negligible multiple scattering is negligible or refer to previous literature.*

   Please see our detailed response to reviewer #1 on this same point.

4. *Line 4-7 page 4: I am not fully convinced by this maximum differential absorption from particulate extinction of 0.01 dB/km. I haven't tried a specific computation but liquid cloud extinction is proportional to $1/\lambda$. So (assuming that the changes in refractive indices are negligible) a change of roughly $3\%$ in lambda should correspond to a change of 38 dB/km/(g/m$^3$), which means that a deep cumulus cloud with 3 g/m$^3$ could produce 0.08 dB/km (an order of magnitude larger than quoted).*

   The issue of differential absorption from hydrometeors in an important one. We have added an extended discussion to the manuscript with Mie calculations for a wide range of realistic cloud and precipitation scenarios.

5. *Line 10 page 5: What is the rationale for using a $\Delta F_{chirp}$ of 60 MHz and thus a range resolution of 2.5 m (with the obvious need of averaging later on for improving the SNR?)? Why not using a smaller bandwidth in first place?*

   Please see our response to reviewer #1, item #5.

6. *Line 11-12 page 7: I do not see the need of dropping the v subscript on, I would recommend to keep it for clarity (otherwise the reader may think it is the total extinction).*

   We have inserted the subscript $v$ on $\kappa$ throughout the manuscript.

7. *Generally in literature SNR values are stated in dB. In Fig.4 and its discussion you use linear units. Fig6 is also confusing to me, why using an obscure value like eta in the x-axis instead of using the SNR itself?*

   We have modified figures 3 and 4 and the associated discussion to express SNR in dB. While the quantity $\eta$ may seem obscure, we feel it is necessary to combine the uncertainty of all the power measurements involved in the humidity extraction for the purposes of the analysis in this section. Since there are $N_f$ different power measurements from 2 different ranges involved, all with different SNR values, the humidity extraction cannot simply be labeled by a single SNR. Due to the confusion caused by this figure, we have decided to remove it from the manuscript.

8. *Fig5: it could help the reader to have a double y axis with the plot of the relative humidity and its uncertainty as well.*

   Since we do not have measurements of the coincident temperature field, we do not feel it is appropriate to report relative humidity values and errors.

9. *Fig5: A couple of points at low and far ranges from the two independent datasets in the bottom right panel seem to disagree, any comment?*

   One shouldn't expect independent sample sets of the same random variable to always have error bars overlapping, as this would signify an overestimate of the distribution variance. The error bars appear representative of the variability between data sets.

10. *Fig 6: apart from the selection of the x-axis I struggle in extracting information from this figure. Why not doing a contour plot of $\sigma_\rho/\rho$ using SNR vs rho e.g. for*

*100 200 and 400m integration? Anyhow I would ask the authors to try to rethink
the figure and present it in a more understandable way.*

We have decided to remove this figure from the manuscript, as well as section
4. Instead, we briefly discuss the trade-off between humidity measurement pre-
cision and resolution at the end of section 3.

---

## Author Comment (AC3) · 23 Oct 2018

Dear Professor Mace,

Thank you for your comments and suggestions regarding our manuscript. Listed below are our itemized responses, with the original comment/question displayed in italics.

1. *At what point will multiple scattering become a limiting factor? At these high frequencies and the typical optical depths of shallow cumulus - perhaps with co-existing precipitation - it seems that multiple scattering may be an issue.*

   Please see our detailed response to reviewer #1 on this same point.

[Figure]

2. *Will the accuracy be sufficient to measure realistic supersaturations in cumulus updrafts? It seems that from a science perspective such knowledge is key. Then combining this instrument with more traditional radars and lidars, one could examine aerosol cloud interaction problems by knowing the cloud droplet number concentration and humidity near cloud base where aerosol populations become activated. Additional science applications could examining the entrainment processes near cloud top where dry tropospheric air is mixed into the marine boundary layer. It seems as though the accuracy required for these topics might push the limitations of the technology.*

These questions regarding the impact of DAR humidity measurement accuracy on science applications are very important, and will be the subject of future focused study after performing validation measurements with coincident measurements from radiosondes, water vapor DIALs, etc. We will provide some useful numbers here which will lay out the expected *precision* of our system, but must leave the *accuracy* discussion for future work. The relative error in the DAR humidity measurement for a two-frequency system in the high-SNR regime is given by $\sigma_\rho/\rho = \xi(N_b)/(\Delta\tau\sqrt{N_p N_b})$, where all of these parameters are defined in the manuscript. As a specific example, we imagine measuring convective updrafts from a ground-based platform, and allow the DAR system to measure for 1 minute. In this case, for 2 transmit frequencies and 1 ms pulse duration, we acquire $N_p = 3 \times 10^4$ pulses, and then use the same downsampling that is used in the paper, $N_b = 11$. For typical boundary-layer parameters, the differential absorption cross section between 174.8 and 167 GHz is $\Delta\kappa = 0.06$ km$^{-1}$/(g/m$^3$). Therefore, using the same retrieval step size as in the paper $R = 200$ m, we find a humidity precision of

$$\frac{\sigma_\rho}{\rho} = 0.19 \times \frac{1\,\text{g/m}^3}{\rho}. \tag{1}$$

If we're observing a level within the updraft with a temperature of 20 °C, the corresponding saturated water vapor density is 17 g/m$^3$. Thus, for 10% supersat-

uration we would have a measurement precision of 1%. At $0\,^\circ$C, the saturated density is 5 g/m$^3$, making the expected precision 4%. These calculations show that it is possible to achieve the necessary precision to measure large supersaturations (10-20%) in intense updrafts. Such supersaturation values are predicted in models that include prognostic supersaturation, but have yet to be observed. However, for ordinary convection, including shallow convection, where supersaturation does not typically exceed 1%, a DAR measurement confirming supersaturation would not be possible. Additional considerations for such a measurement include the necessary retrieval resolution and the timescale over which an initially supersaturated volume becomes one with RH $\leq$ 100 (e.g. from advection).

Additionally, it is important to point out that with more freedom to transmit in other frequency bands near the 183 GHz line, the retrieval resolution and precision can be substantially reduced. It is easy to find a pair of frequencies for which $\Delta\kappa$ is an order of magnitude larger or more. If such frequencies were used, one can use that factor of 10 increase in sensitivity to reduce the step size $R$, the humidity precision, or a combination of both.

It is not exactly clear to us what is meant by the final sentence as it pertains to dry-air entrainment near cloud tops. As an example measurement for this phenomenon, imagine an airborne DAR flying just above the marine boundary layer. From the first radar echos (in range) one can measure the short water column between the aircraft and the cloud top. Then, the in-cloud humidity profile can be retrieved using the subsequent radar returns from throughout the cloud volume. We would expect the dry-air entrainment signature to be a sharp gradient of the water vapor density collocated with the cloud top inferred from the reflectivity profile. With the current DAR system (i.e. in the 167 to 174.8 GHz band), the retrieval resolution is too coarse to resolve this mixing process at the top of the stratocumulus layer, which has a spatial scale on the order of 10 meters. As mentioned in the previous paragraph, a DAR operating closer to the line center

achieves much higher spatial resolution, and could potenitally resolve this effect.

3. *For the topics identified in point 2, validation with radiosondes would be inadequate. Would current in situ technology for measuring water vapor allow for validation of the technique?*

4. We're not quite certain what is intended by this comment and question. In general, our validation approach will be to utilize radiosonde measurements in scenarios where they measure RH and T very accurately, and compare this with the DAR measurements. Note that in typical cloud scenarios, RH $\approx 100\%$ and is measured well by radiosondes, with the resolution, precision and accuracy exceeding that expected for the DAR. Since there is nothing fundamentally different from a millimeter-wave scattering perspective between the proposed validation scenario and those referenced in point 2, we see no need for improved validation data.

---

## Author Response (AR2)

Dear Associate Editor,

We are pleased to hear that our manuscript is accepted subject to minor revisions. Thank you for your comments regarding our manuscript. Listed below are our itemized responses.

1. *The first is in sentence 1 of paragraph 2 of the introduction; to "complement the weakness of existing methods" logically (or grammatically) means that you add another weakness. You should simply delete the words "weakness of".*

   This phrase has been removed in the edited final manuscript.

2. *Second, in figure 4 you need to add the units for the legend in panel (a). This could be done in the figure itself or in the caption.*

   We assume that you are referring to figure 2, since figure 4(a) does not have a legend. The caption for figure 2 has been edited to give the units of these numbers.

3. *Third, in the new last paragraph of section 2.1, it is somewhat clumsy to use an abbreviation (LWC) as a symbol in the equation, because it is unclear in first reading where the symbol begins and ends. If you don't want to introduce a simpler symbol you could use a "times" symbol before and after LWC.*

   We have replaced the abbreviation LWC with the variable $\mathcal{L}$.

4. *Fourth, in the second paragraph of section 2.2, you need to clarify what you mean by "restricted". If this is in the regulatory sense you should say so explicitly, as otherwise this sentence will be contradictory to all but specialists.*

   We first note that in the first paragraph of the introduction, we have already provided a brief discussion of the international regulations that affect these frequency bands. Additionally, in the edited final manuscript we have added a clarifying phrase in the second paragraph of section 2.2.

The following is a list of all relevant changes made to the manuscript:

1. Removed phrase "weaknesses of" in section 1, paragraph 2.

2. Replaced variable "LWC" with $\mathcal{L}$ for representing liquid water content throughout section 2.1.

3. Added a sentence in the caption to figure 2 clarifying the units of the numbers in the legend in panel (a).

5  4. Added phrase clarifying what is meant by restriction in section 2.2, paragraph 2.

[revised manuscript text omitted]